# Bioactive PKS–NRPS Alkaloids from the Plant-Derived Endophytic Fungus *Xylaria arbuscula*

**DOI:** 10.3390/molecules27010136

**Published:** 2021-12-27

**Authors:** Ya Wang, Sinan Zhao, Tao Guo, Li Li, Tantan Li, Anqi Wang, Dandan Zhang, Yanlei Wang, Yi Sun

**Affiliations:** 1School of Life Science and Engineering, Lanhzou University of Technology, Lanzhou 730050, China; wangya502@163.com (Y.W.); 18530869018@163.com (S.Z.); 2Institute of Chinese Materia Medica, China Academy of Chinese Medical Sciences, Beijing 100010, China; tantanna0309@163.com (T.L.); mickey-wang@139.com (A.W.); z2531817909@163.com (D.Z.); W15739539626@163.com (Y.W.); 3School of Pharmacy, Henan University of Chinese Medicine, Zhengzhou 450046, China; gt010010@hactcm.edu.cn; 4Institute of Materia Medica, Chinese Academy of Medical Sciences & Peking Union Medical College, Beijing 100010, China; annaleelin@imm.ac.cn

**Keywords:** *Xylaria arbuscula*, PKS–NRPS hybrids, cytotoxicity, anti-inflammatory activity

## Abstract

A novel hybrid PKS–NRPS alkaloid, xylarialoid A (**1**), containing a 13-membered macrocyclic moiety and [5,5,6] fused tricarbocyclic rings, together with ten known cytochalasins (**2**–**11**), was isolated from a plant-derived endophytic fungus, *Xylaria arbuscula*. The chemical structures of all compounds were elucidated using 1D and 2D NMR, HR ESIMS spectroscopic analyses, and electronic circular dichroism (ECD) calculation. Compounds **1**–**3** and **10** exhibited significant antitumor activities against A549 and Hep G2 cell lines, with IC_50_ values of 3.6–19.6 μM. In addition, compound **1** showed potent anti-inflammatory activity against LPS-induced nitric oxide (NO) production in macrophage RAW 264.7 cells (IC_50_, 6.6 μM).

## 1. Introduction

The *Xylaria* genus belongs to the family of Xylariaceae, which is commonly found among saprophytic and endophytic fungi [1]. Several studies have proved that the genus *Xylaria* can produce a variety of bioactive secondary metabolites including terpenes, cytochalasins, alkaloids, polyketides, etc. [2,3]. Many of these possess promising biological activities associated with drug discovery, such as cytotoxicity and antimalarial and antimicrobial activities [4]. In the past ten years, a class of polyketide synthase–nonribosomal peptide synthetase (PKS–NRPS) alkaloids has attracted the attention of scientists due to their unique structure. These alkaloids have 12- or 13-membered ring systems, γ-lactams, and fused [6,5,6] tricarbocyclic or [6,5,6,5] tetracarbocyclic cores, which were isolated from different fungal species and showed antitumor, antifungal, and antituberculosis biological activities [5,6,7,8,9,10]. The biosynthesis pathway of the PKS–NRPS hybrid alkaloids was through P450-catalyzed cyclization to form paracyclophane, and all of the ring system was constructed from a PK–NRP product via oxidation [11]. Cytochalasins are a class of PKS–NRPS hybrids and have been isolated from the fungi of *Chaetomium, Xylaria, Rosellinia,* and *Zygosporium* [12,13,14,15]. They have tricyclic core structures, which are in turn each composed of a perhydroisoindole moiety and a macrocyclic ring. Some of the cytochalasins are phytotoxins or virulence factors, so they possess cytotoxic activities [16,17,18]. In addition, cytochalasins are biosynthesized through a hybrid PKS–NRPS pathway in which Knoevenagel condensation is a key step during the biosynthesis [19]. During our continuous studies on antitumor secondary metabolites from endophytic fungi, we screened a fungus, *Xylaria arbuscula*, isolated from the medicinal plant *Rauvolfia vomitoria* by using LC–MS analysis and bioactivity-guided methods. Its EtOAc extract was subjected to repeated chromatographic purification to afford a novel hybrid PKS–NRPS alkaloid, named xylarialoid A (**1**), and ten cytochalasins (**2**–**11**) (Figure 1). All eleven compounds were evaluated for their cytotoxic activities against A549 and Hep G2 cell lines and their inhibitory activities toward LPS-induced NO production in macrophages. Herein, we report the isolation and structural elucidation of compound **1** and the anti-inflammatory and cytotoxic activities of compounds **1**–**11**.

## 2. Results and Discussion

### 2.1. Structural Elucidation of Compound **1**

Compound **1** was obtained as a light yellow solid. Its molecular formula was determined to be C_29_H_33_NO_5_ by HR ESIMS [M + H] ^+^
*m/z* 476.2437, indicating 14 degrees of unsaturation. The ^1^H NMR (Table 1 and Appendix A) spectrum indicated the presence of a NH proton signal at *δ*_H_ 8.30 (d, *J* = 2.2 Hz, 1H); eight olefinic/aromatic protons, including the aromatic proton signals at δ_H_ 7.27 (dd, *J* = 8.3, 2.1 Hz), 6.99 (dd, *J* = 8.2, 2.5 Hz), 6.90 (d, *J* = 2.2 Hz), and 6.92 (dd, *J* = 8.5, 2.0 Hz); a group of terminal olefin protons; and two methyl groups at *δ*_H_ 0.92 (d, *J* = 6.6 Hz) and 1.01 (d, *J* = 6.4 Hz). The ^13^C NMR and HSQC spectra (Table 1 and Appendix A) of **1** showed the presence of a 1,4-disubstituted benzene; a disubstituted double bond; a terminal olefin; two methyls; three methylenes; eight methines, including two oxygenated methines; two carbonyl carbons; and an oxygenated quaternary carbon. The planar structure of compound **1** was deduced by HMBC and ^1^H–^1^H COSY spectra (Figure 2 and Appendix A). The HMBC correlations from H-2 to C-1, C-3, and C-15, from H-3 to C-5 and C-14, from H-6 to C-4, C-8, C-12, and C-13, from H-8 to C-10 and C-12, and from H-13 to C-6 and C-7, combined with the observed ^1^H–^1^H COSY correlations of H-1/H-2/H-3, H-3/H-4/H-5, H-5/H-6/H-7, H-7/H-8/H-9, H-9/H-10/H-11, H-11/H-12/H-13, H-13/H-14/H-15, H-9/H_3_-19, and H-11/H_3_-20, indicated the presence of a decahydrofluorene moiety attached to a vinyl group at C-3 and two methyls at C-9 and C-11. The HMBC correlations from NH to C-1′, C-2′, and C-17 and from H-1′ to C-2′ and C-18 suggested the presence of a *γ*-lactam system. Additional ^1^H-^1^H COSY correlations of H-5′/H-6′ and H-8′/H-9′ and key HMBC correlations from H_2_-3′ to C-2′, C-4′, C-5′, and C-9′ and from H-1′ to C-3′ revealed the linkage between a *para*-substituted phenethyl moiety and a *γ*-lactam ring. Further HMBC correlations from H-1′ to C-17, C-18, and C-2′, as well as the chemical shifts of C-17, C-1′, and C-2′, indicated the presence of an epoxide moiety at C-1′ and C-17 in the *γ*-lactam moiety and a hydroxy at C-2′. Moreover, the HMBC corrections from H-3 and H-15 to C-16 showed that the *γ*-lactam and decahydrofluorene system were connected through a ketone carbonyl group. Finally, the HMBC correlation from H-13 to C-7′ confirmed the presence of a 13-membered ring and completed the whole planar structure of **1**.

The relative configuration of **1** was determined by the ROESY spectrum (Appendix A) and the coupling constants in the ^1^H NMR (Figure 2). The observed ROESY cross-peaks of H-7 with H-14 and H-14 with H-3 suggested that they were cofacial and on the β-orientation. However, the vicinal coupling constant of *J*_9,10α_ = 12.2 Hz and the NOE correlations of H-10α with H_3_-9/H_3_-11 revealed that these protons were on the same side, and they were assigned to be α-oriented. The ROESY cross-peaks of H-13 with H-6, H-12, H-15, and H-6′; H-15 with H-6; and H-1′ with H-5′ and the amide group (NH) indicated that they were on the α-orientation and that the benzene ring was rotation restricted. Additionally, 1D NOE difference experiments (Appendix A) were performed to identify the important spatial interactions of **1**. The signals of H-13 and H-6 were enhanced when H-15 at *δ*_H_ 2.37 was irradiated. Furthermore, the coupling constant between H-12 and H-13 (*J*_12,13_ = 8.5 Hz) confirmed their *trans*-configuration, and H-13 was axial. An additional coupling constant (*J* = 9.8 Hz) between H-4 and H-5 assigned the Δ^4,5^ geometry as *Z*. The configurations of the fused [5,6,6] tricarbocyclic ring was similar to that of trichobamide A [20]. In order to determine the absolute configuration of **1**, the time-dependent density functional theory (TDDFT) was used to calculate the theoretical electronic circular dichroism (ECD) spectrum (Figure 3). At the B3LYP level, the enantiomer configurations of **1a** and **1b** were compared by ECD calculation. The experimental ECD curve of **1** was in consistent with the calculated curve of **1a** (Figure 3). Therefore, the absolute structure of **1** was finally established as 3*S*, 6*R*, 7*S*, 9*R*, 11*S*, 12*R*, 13*S*, 14*S*, 15*R*, 17*R*, 1′*R*, 2′*R*, and **1** was named xylarialoid A.

### 2.2. Cytotoxic and Anti-Inflammatory Activities

The cytotoxic activities of compounds **1**–**11** against A549 and HepG2 were evaluated using the MTT method (Table 2). Compounds **1**–**3** and **10** exhibited potent cytotoxicity against the A549 and Hep G2 cell lines, with IC_50_ values between 3.6 and 19.6 μM. Additionally, the anti-inflammatory effects of compounds **1**–**11** were explored against NO production in LPS-induced RAW264.7 macrophage cells. Among the test compounds, only compound **1** showed the potent inhibitory activity, with an IC50 value of 6.6 μM. The cytochalasins did not display anti-inflammatory activity (≥50 μM).

## 3. Materials and Methods

### 3.1. General Experimental Procedures

Optical rotations were measured using a PerkinElmer 241 polarimeter. NMR spectra were recorded on a Bruker ARX-600 spectrometer operating at 600 MHz for ^1^H and 150 MHz for ^13^C, using DMSO-*d*_6_ (*δ*_H_ 2.50 and *δ*_C_ 39.50) as residual solvent signals for reference. High-resolution electrospray ionization mass spectrometry (HRESIMS) data were acquired on a Waters Vion QTOF/MS spectrometer (Waters Mocromass, Manchester, UK) in positive electrospray ionization mode. An UPLC reversed phase C18 analytical column (2.1 mm × 100 mm, 1.7 μm, BEH, Waters) was adopted. High-performance liquid chromatography (HPLC) was performed on an Agilent 1260 HPLC system with a UV detector (Angilent, Technologies Co., Ltd., Palo Alto, CA, USA) coupled with analytical or semipreparative Cosmosil 5C_18_-ARII columns (4.6 mm × 250 mm and 250 mm × 10 mm) and Cosmosil 5C_18_-MSII (250 mm × 4.6 mm and 250 mm × 10 mm). Thin-layer chromatography (TLC) was performed on GF_254_ plates precoated with silica gel (Qingdao Haiyang Chemical Co., Ltd., Qingdao, China). Column chromatography (CC) was performed on a Sephadex LH-20 (Pharmacia Fine Chemical Co., Ltd., Uppsala, Sweden), silica gel (200–300 mesh, Qingdao Ocean Chemical Factory), and octadecyl-functionalized silica gel (ODS) (50 μm, YMC Japan). Human lung cancer cells A549 and human hepatocellular carcinoma cells HepG2 were purchased from Procell Life Science & Technology Co., Ltd. Mouse monocyte macrophage Raw 264.7 cells were purchased from the Chinese National Infrastructure of Cell Line Resource (NICR).

### 3.2. Fungal Material

The fungal strain was isolated from the leaves of the plant *Rauvolfia vomitoria* belonging to the apocynaceae family, which was collected from Yunnan Province, China, in July 2013. The plant species was identified by Dr. Yi Sun. The fungus was identified as *Xylaria arbuscula* by analyses of its rRNA gene sequences (GenBank accession number KY951913.1, see Appendix A). The strain, named cuiluo-leaf-1p-e, is currently deposited in the institute of Chinese Materia Medica, China Academy of Chinese Medical Sciences.

### 3.3. Fermentation and Extraction

The fungal strain was cultured on a plate of potato dextrose agar (PDA) at 27.0 ± 0.5 °C for 3 days, which was then inoculated into 110 × 0.5 L Erlenmeyer flasks each containing 40 g sterilized rice and 60 mL water (sterilization at 121 °C for 20 min).

After incubation at 28 ± 0.5 °C for 9 days, the fermented rice media was exhaustively extracted with ethyl acetate by ultrasonication three times, and the organic solvent was evaporated to dryness under reduced pressure to obtain the crude extract. Then, the EtOAc layer was dissolved in methanol and extracted with petroleum ether (1:1) to remove grease from the rice. After extraction, 5.0 g of crude extract was yielded.

### 3.4. Isolation and Purification

The crude extract (5.0 g) was fractionated on a flash ODS column eluted successively with 2 L each of MeOH–H_2_O (40:60, 60:40, 80:20, 100:0) to yield four fractions. The fraction eluted with MeOH–H_2_O (80:20) was subjected to a silica gel column by step gradient elution with CH_2_Cl_2_/Methanol (from 50:1 to 0:1) and yielded four fractions (Frs. A–D). Fr.A was purified by semipreparative HPLC (Cosmosil 5C_18_-MSII, 250 × 10 mm i.d., 5 μm, 3 mL/min) with gradient elution from 50 to 75% acetonitrile in H_2_O with 0.2% AcOH to afford compounds **6** and **7** (**6**, t_R_ = 25.0 min, 2.1 mg; **7**, t_R_ = 32.0 min, 1 mg). Fr.B was purified by HPLC (Cosmosil 5C_18_-MSII, 250 × 10 mm i.d., 5 μm, 3 mL/min) with gradient elution from 60 to 75% acetonitrile in H_2_O with 0.2% AcOH to afford compounds **8** and **9** (**8**, t_R_ = 22.0 min, 4 mg; **9**, t_R_ = 37.0 min, 3 mg). Fr.C was purified by HPLC (Cosmosil 5C_18_-MSII, 250 × 10 mm i.d., 5 μm, 3mL/min) with gradient elution from 50 to 85% acetonitrile in H_2_O with 0.2% AcOH to afford compounds **1**, **10,** and **11** (**1**, t_R_ = 35.0 min, 1.8 mg; **10**, t_R_ = 23.0 min, 2.5 mg; **11**, t_R_ = 35.0 min, 2.8 mg). The fraction eluted with 60% MeOH was subjected to a Sephadex LH-20 (CH_2_Cl_2_-MeOH, 1:1) to yield five subfractions (A–E). Fr.B was then isolated by silica gel column chromatography (CC) (200–300 mesh), eluting with a CH_2_Cl_2_–acetone gradient system (from 50:1 to 1:1) to yield five subfractions (Frs.1–5) monitored by TLC. Fr.2 was further purified by semipreparative HPLC (Cosmosil 5C_18_-ARII columns, 250 × 10 mm i.d., 5 μm, 3 mL/min) with a gradient elution of acetonitrile/H_2_O (45/55%) to yield compounds **2** and **3** (**2**, t_R_ = 20.0 min, 2.1 mg; **3**, t_R_ = 28.0 min, 1.7 mg). Fr.3 was purified by HPLC (Cosmosil 5C_18_-ARII column, 250 × 10 mm i.d., 5 μm, 3 mL/min) with a gradient elution from 50 to 65% acetonitrile in H_2_O with 0.2% AcOH to yield compounds **4** and **5** (**4**, t_R_ = 20.0 min, 10.2 mg; **5**, t_R_ = 35.0 min, 6.0 mg). 

Compound **1**: shallow yellow solid; [α]^25^_D_ −20.5 (c 0.10, MeOH); CD (MeOH) 196 (Δε +1.33) nm, 232 (Δε –2.78) nm. ^1^H NMR (600 MHz, DMSO-*d_6_*) and ^13^C NMR (150 MHz, DMSO-*d_6_*) data (Table 1). ESIMS *m/z* 476 [M + H] ^+^; HR-ESIMS *m/z* 476.24368 [M + H] ^+^, calculated for C_29_H_33_NO_5_, 476.24368.

### 3.5. Computation of ECD

A conformational search was carried out in the MMFF94 molecular mechanics force field using the MOE software package, and all the conformers within an energy window of 10 kcal/mol were regarded as the initial conformations. The geometry optimization and frequency calculations were performed with Gaussian16 RevB.01, using the ωB97XD or B3LYP functional at the 6-311G(d,p) level of theory to verify the stability and obtain the energies at 298.15 K and 1 atm pressure. The Boltzmann distribution was calculated according to Gibbs free energies of the conformations. ECD calculations were conducted by using the Cam-B3LYP functional at the TZVP level of theory. The Solvation Model Based on Density (SMD) was used as the solvation model. The Boltzmann-averaged ECD spectra were obtained by using SpecDis 1.71 software [21].

### 3.6. Cytotoxicity Assay

The cytotoxic activities of **1–11** against human lung cancer cells A549 and human hepatocellular carcinoma cells HepG2 were determined by the 3-(4,5-dimethyl-2-thiazolyl)-2,5-diphenyl-2-H-tetrazolium bromide (MTT) assay [22,23]**.** The cells were cultured in RPMI-1640 containing 10% (*v*/*v*) fetal bovine serum (FBS) and 0.4% (*v*/*v*) penicillin–streptomycin solution (10,000 units/mL penicillin and 10,000 μg/mL streptomycin, 100×) at 37 °C under 5% CO_2_. The cells were digested by trypsinization and then diluted to a concentration of 1 × 10^4^ cells/mL. The diluted cell suspensions were then added into 96-well microtiter plates (200 μL per well) and incubated in a CO_2_ incubator at 37 °C for 24 h. After 24 h, 2 μL of test samples was added to each plate well and incubated for 72 h. Adriamycin was used as the positive control, and the blank control contained 2 μL CH_3_OH. After incubation, MTT solution was added, and the plates were incubated for 4 h. The supernatant liquid was removed, and the cells were disrupted with 150 μL of DMSO for 10 min. The absorption was measured at 570 nm.

### 3.7. Anti-Inflammatory Assay

The anti-inflammatory activities of compounds **1**–**11** were evaluated on lipopolysaccharide-induced murine macrophage cell lines (RAW 264.7) [24,25]. RAW 264.7 cells were cultured in DMEM high glucose containing 10% FBS at 37 °C under 5% CO_2_ until the cells were confluent. For experiment, 2–5 generations of cultured cells were used. The procedure was as follows. First, raw 264.7 cells in logarithmic growth phase were inoculated into 24-well plates at a density of 2 × 10^5^ cells·ml^−1^ (1 mL per well) and incubated in a CO_2_ incubator at 37 °C for 24 h. After 24 h, RAW 264.7 cells were induced to inflammation using 1 mg/mL lipopolysaccharide. Second, the Griess method was used to detect the release of NO in cell supernatant. The diluted cell suspensions were added to 96-well plates. To each well, 0, 0.5, 1, 1.5, 2, 4, 6, and 8 μL of NaNO_2_ standard solution (1 mol/L) were successively added. To the 96-well plates was added 100 μL of the obtained supernatant. The 96-well microtiter plates were shaken, 100 μL Griess reagent was added into each well, and the absorption was measured at 570 nm. Resveratrol was used as the positive control; DMSO was used as the blank control.

## 4. Conclusions

In summary, a novel PKS–NRPS alkaloid, xylarialoid A (**1**), together with ten known cytochalasins (**2–11**), was isolated from a culture of *Xylaria arbuscula*. The structure of compound **1** was determined by comprehensive NMR, HR ESIMS, and ECD spectroscopic data. Xylarialoid A was found to have a rare tyrosine–decahydrofluorene skeleton with a fused 6/5/6 tricarbocyclic core and a 13-membered *para*-cyclophane ring system. Xylarialoid A displayed both notable cytotoxicity against A549 and Hep G2 tumor cell lines and potent inhibition of nitric oxide (NO) production in LPS-activated RAW 264.7 macrophages. Meanwhile, compounds **2**, **3**, and **10** exhibited significant cytotoxicities against A549 and Hep G2 cell lines (IC_50_, 3.6~19.6 μM).

## Figures and Tables

**Figure 1 molecules-27-00136-f001:**
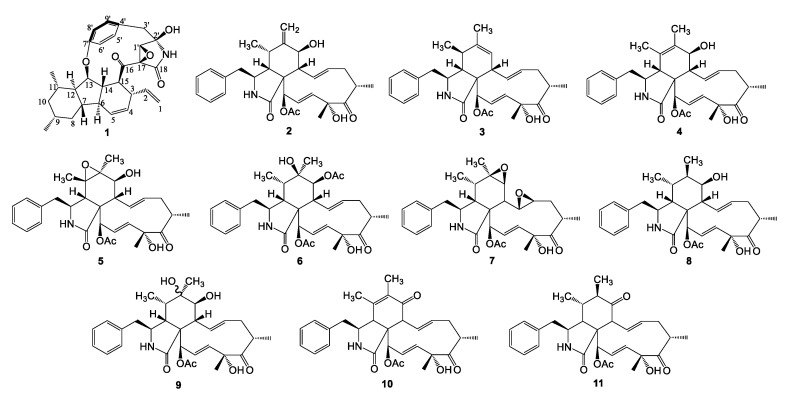
Chemical structures of compounds **1–11** from *Xylaria arbuscula*.

**Figure 2 molecules-27-00136-f002:**
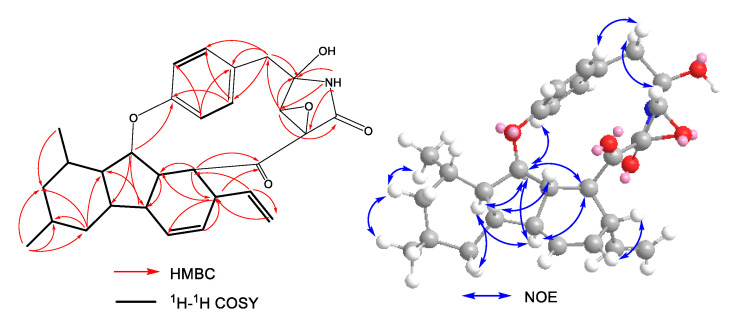
^1^H–^1^H COSY, key HMBC, and NOESY correlations of compound **1.**

**Figure 3 molecules-27-00136-f003:**
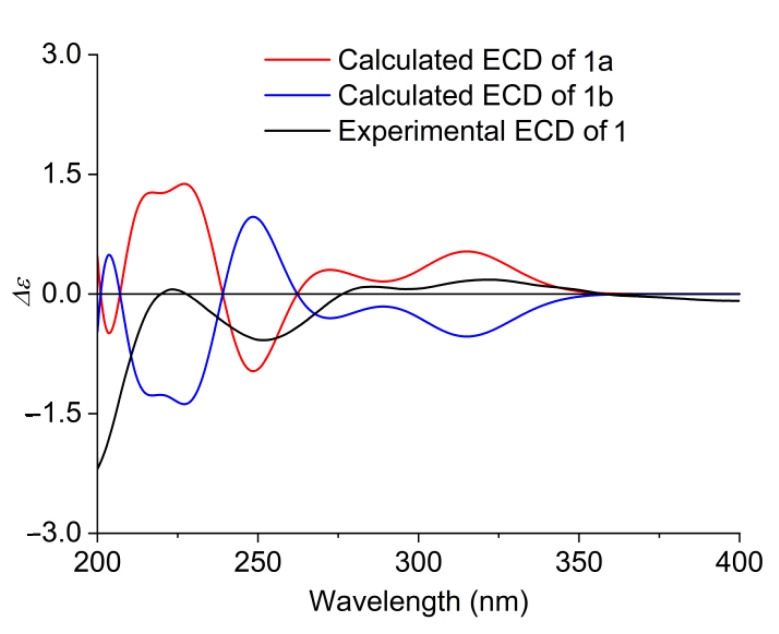
Experimental and calculated ECD of **1** (**1a, 1b**) in MeOH.

**Table 1 molecules-27-00136-t001:** ^1^H and ^13^C NMR data for compound **1** in DMSO-*d_6._*

Position	*δ*_H_ (*J*, Hz)	*δ*c (ppm)
1	4.88 (dd, 10.1, 1.9), 4.83 (m)	116.6
2	5.26 (m)	137.1
3	2.88 (m)	42.9
4	5.08 (dd, 9.8, 3.5)	127.1
5	5.95 (d, 9.8)	130.1
6	2.77 (m)	38.4
7	2.12 (m)	41.3
8	1.90 (m), 0.76 (dd, 11.9, 11.9)	36.8
9	1.57 (m)	33.6
10	1.74 (m), 0.66 (dd, 12.2, 12.2)	44.8
11	1.64 (m)	32.1
12	1.06 (m)	59.3
13	4.83 (dd, 8.5, 6.3)	79.1
14	1.72 (m)	46.8
15	2.37 (dd, 11.1, 6.7)	48.9
16	—	200.5
17	—	64.8
18	—	165.3
19	0.92 (d, 6.6)	22.5
20	1.01 (d, 6.4)	19.6
1′	3.41 (m)	59.8
2′	—	82.9
3′	3.14 (d, 13.2), 3.01 (d, 13.2)	45.3
4′	—	129.3
5′	7.27 (dd, 8.3, 2.1)	133.1
6′	6.99 (dd, 8.3, 2.2)	122.4
7′	—	157.8
8′	6.90 (dd, 8.5, 2.2)	120.9
9′	6.92 (dd, 8.5, 2.1)	128.6
NH	8.30 (d, 2.2)	—
OH	3.64 (d, 2.2)	—

**Table 2 molecules-27-00136-t002:** IC_50_ values (μM) of the cytotoxic activity of compounds (**1**–**11**).

Cells	IC_50_ (μM) Values of Compounds					
	1	2	3	4	5	6
A549	14.6 ± 0.1	19.6 ± 0.2	3.6 ± 0.2	—^b^	—^b^	—^b^
Hep G2	15.3 ± 0.2	15.2 ± 0.3	9.0 ± 0.3	30.3 ± 0.2	—^b^	—^b^
	**7**	**8**	**9**	**10**	**11**	adriamycin ^a^
A549	—^b^	—^b^	—^b^	15.7 ± 0.3	24.8 ± 0.2	3.2 ± 0.2
Hep G2	—^b^	32.0 ± 0.2	—^b^	17.5 ± 0.4	15.1 ± 0.3	4.4 ± 0.3

^a^ Adriamycin and DMSO were used as positive and negative controls, and the data were expressed as the means ± SD (*n* = 3); ^b^ IC_50_ values were more than 40 μM.

## Data Availability

Not applicable.

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
