# Peer review of "Bioactive PKS–NRPS Alkaloids from the Plant-Derived Endophytic Fungus Xylaria arbuscula"

_molecules, 2021, doi:10.3390/molecules27010136_

Round 1

Reviewer 1 Report

The manuscript of Yi Sun et al. describes  the isolation of 11 compounds from Xylaria arbuscula. With the exception of one substance (1) all compounds are known. The unknown compound 1 was structurally characterized using NMR, MS and CD-methods with special emphasis on the NMR-spectroscopic elucidation  of the relative configuration of the new natural product. For me this part needs clarification:

1) Minor points: The signal of the NH-proton is not a doublett as stated in section 2.1 and visible in the spectrum. Is there really a COSY crosspeak from H-1 to H-15 (5J-coupling)? The Δ4,5-geometry is Z not E as given in the same section.

Major points:

2) How did the authors assign the diastereotopic protons at C-10, C-8 and C-3'. Especially the assignment at C-10 is of interest here, because one of the protons (H-10α) has been used to interpret NOE-connectivities (danger of circular arguments in case this assignment was also based on NOEs!)

3) The interpretation of the NOESY data was only a qualitative one (crosspeak is there/crosspeak is absent). This may be problematic especially in case of spin-diffusion which may occur when the mixing time was too long. No information on this issue is given in the manuscript.

4) If one compares the proposed relative configuration of 1 with structurally related compounds described in the literature and mentioned by the authors themselves, it is interesting to note that 1 obviously has a hitherto undescribed new relative configuration if one considers C9/C11/C13/C15 and C3 as given in the table below:

rel. config. 1 GKK1032s Pyrrocidines Hirsutellones
11,13 trans trans trans - (no methyl)
13,15 trans cis cis trans
15,3 cis cis trans cis
3,13 trans cis trans trans
9,13 trans trans trans cis

This very interesting point was not discussed by the authors. It may have impact on the interpretation of the biogenesis of the compounds and should be discussed.

Apart from these issues the work surely merits publication, provided that:

The authors include quantitative NOE-data in the supporting information (cross-peak volumes above and below the diagonal) with information on measuring conditions, especially the mixing time(s) used.

Author Response

Dear Editor and Reviewers,

Thank you very much for taking time to review our manuscript. We really appreciate all your generous comments. We have revised the manuscript according to your suggestions, and marked the corrections using the blue highlights. Please check the responses as follows.

Response to Reviewer 1

Minor points:

  • The signal of the NH-proton is not a doublett as stated in section 2.1 and visible in the spectrum. Is there really a COSY crosspeak from H-1 to H-15 (5J-coupling)? The Δ4,5-geometry is Z not E as given in the same section.

Response: Thanks for your suggestions. We have checked the 1H NMR spectrum of compound 1, and found the signal of the NH-proton is a small doublet (J = 2.2 Hz). As shown in the following figure. In addition, the geometry of the double bond at C-4 and C-5 was Z. We have revised it.

Major points:

  • How did the authors assign the diastereotopic protons at C-10, C-8 and C-3'. Especially the assignment at C-10 is of interest here, because one of the protons (H-10α) has been used to interpret NOE-connectivities (danger of circular arguments in case this assignment was also based on NOEs!)

Response: Thanks for your comment. We calculated the coupling constants of H-10α at δ 0.66 (J = 11.9 Hz), indicating that H-10α should be axial, and H-10α showed NOE correlation with H-12. Thus, H-10α and H-12 were on the same side of α-orientated.

  • The interpretation of the NOESY data was only a qualitative one (crosspeak is there/crosspeak is absent). This may be problematic especially in case of spin-diffusion which may occur when the mixing time was too long. No information on this issue is given in the manuscript.

Response: Thanks for your suggestion. The two-dimensional NOE correlation in this experiment is qualitative but not quantitative. We checked the measurement parameters of ROESY experiment (not NOESY) as follows. The ROESY pulse sequence is roesyphpp: 2, spinlock times: 600 ms, and the Relaxation Delay: 1.2. To confirm the relative configuration of compound 1, we also determine 1D NOE experiment. We have supplemented the important 1D NOE results and corrected the relative configuration at C-15 (eg. H-15 with H-6 and H-13) in the manuscript. The spinlock times for one-dimensional NOE irradiation was also 600 ms. Please check the NOESY discussion highlighted in blue in the manuscript.

  • If one compares the proposed relative configuration of 1 with structurally related compounds described in the literature and mentioned by the authors themselves, it is interesting to note that 1 obviously has a hitherto undescribed new relative configuration if one considers C9/C11/C13/C15 and C3 as given in the table below:

This very interesting point was not discussed by the authors. It may have impact on the interpretation of the biogenesis of the compounds and should be discussed.

Response: Thanks for your suggestions. We have re-checked the NOE correlations in the ROESY spectrum and determined 1D NOE spectrum to confirm the relative configurations of 1. The new compound has a similar configuration to that of trichobamide A, which was cited it and discussed in the manuscript.

Reviewer 2 Report

Dear Authors,

The manuscript reported one new natural product and ten known compounds from endophytic Fungus Xylaria arbuscula of plant Rauvolfia vomitoria collected at China. The structure was studied by 2D NMR, MS and DFT-ECD, and all isolated compounds were tested for their NO inhibitory and cytotoxic activities.  

1) SI file, all the figure caption "DMSO-D6" should be "DMSO-d6". All the proton spectra from 10 - 16 ppm have no signals, therefore it might be better to remove this region, so that readers would have a better view on them. Compound 1 has 1.8 mg, NMR measurements were collected using 600 MHz instrument, but HSQC, HMBC and TOCSY seem a bit off to me, did you use Shigemi tube or normal tube (5 mm)? Usually ~2 mg (MW 300-700) with a Shigemi tube on 600 MHz could get a quite good spectra.

2) There is a lot of stuff about cytochalasans, the NPR review [ref 12] of cytochalasans was cited but the latest paper about cytochalasans was not cited, might include it for the readers to catch up what is happening to cytochalasans research direction.

Chemical and Genetic Studies on the Formation of Pyrrolones During the Biosynthesis of Cytochalasans. DOI: 10.1002/chem.202004444

3) The title and text about novel hybrid PKS-NRPS alkaloid (1) is a bit unusual. Don't the core structure of 1 resemble any existing natural products? If no, a name of this chemical skeleton must be given, so that this name can be use to refer 1 instead of hybrid PKS-NRPS.

4) How do Authors determine 1 as PKS-NRPS? The fungal strain has obtained the genome sequence? If yes, please provide the genome sequence accession number. If no, compound 1 cannot be confidently stated as PKS-NRPS. Is there are any similar core skeleton to those of 1?

5) Below are some MDPI journals detailing the cytotoxic and anti-inflammatory assays that might consider to be included in section 3.6-3.7.

Mar. Drugs 201816(4), 99; https://doi.org/10.3390/md16040099

Mar. Drugs 202119(10), 565; https://doi.org/10.3390/md19100565

6) DFT calculation software should be cited.

Author Response

Dear Editor and Reviewers,

Thank you very much for taking time to review our manuscript. We really appreciate all your generous comments. We have revised the manuscript according to your suggestions, and marked the corrections using the blue highlights. Please check the responses as follows.

Response to Reviewer 2

  • SI file, all the figure caption "DMSO-D6" should be "DMSO-d6". All the proton spectra from 10 - 16 ppm have no signals, therefore it might be better to remove this region, so that readers would have a better view on them. Compound 1 has 1.8 mg, NMR measurements were collected using 600 MHz instrument, but HSQC, HMBC and TOCSY seem a bit off to me, did you use Shigemi tube or normal tube (5 mm)? Usually ~2 mg (MW 300-700) with a Shigemi tube on 600 MHz could get a quite good spectra.

Response: Thanks for your suggestion. We have revised all the figure caption of "DMSO-D6" to "DMSO-d6" in the manuscript and SI material. And we re-scale the proton spectra from 0-8.5 ppm in the spectrum. We actually used the normal tube (3mm) for the measurement of 1D and 2D NMR spectra on 600 MHz machine.

Please see in the revised support information.

  • There is a lot of stuff about cytochalasans, the NPR review [ref 12] of cytochalasans was cited but the latest paper about cytochalasans was not cited, might include it for the readers to catch up what is happening to cytochalasans research direction.

Chemical and Genetic Studies on the Formation of Pyrrolones During the Biosynthesis of Cytochalasans. DOI: 10.1002/chem.202004444

Response: Thanks to the reviewer providing the valuable references about cytochalasans. We have cited this reference in the revised manuscript. Also, we added more references of the cytochalasans. Please check the manuscript.

  • The title and text about novel hybrid PKS-NRPS alkaloid (1) is a bit unusual. Don't the core structure of 1 resemble any existing natural products? If no, a name of this chemical skeleton must be given, so that this name can be use to refer 1 instead of hybrid PKS-NRPS.

Response: Thanks for your suggestion. It has reported a few compounds possessing the same skeleton with compound 1, and we have referred some literatures in the manuscript. We have named compound 1 as xylarialoid A.

  • How do Authors determine 1 as PKS-NRPS? The fungal strain has obtained the genome sequence? If yes, please provide the genome sequence accession number. If no, compound 1 cannot be confidently stated as PKS-NRPS. Is there are any similar core skeleton to those of 1?

Response: Thanks for your suggestion. We have added the strain’s information, which its GenBank accession number is KY951913.1), and Xylaria arbuscula contains PKS-NRPS gene cluster. We also have found the compounds with the same skeleton from different fungal sources.

  • Below are some MDPI journals detailing the cytotoxic and anti-inflammatory assays that might consider to be included in section 3.6-3.7.

Mar. Drugs 2018, 16(4), 99; https://doi.org/10.3390/md16040099

Mar. Drugs 2021, 19(10), 565; https://doi.org/10.3390/md19100565

Response: Thanks to the reviewer provide valuable references about cytochalasans. We have cited these references in the revised manuscript. Please check in the manuscript.

  • DFT calculation software should be cited.

Response: Thanks for your comment. We have cited DFT calculation software in the revised manuscript.

Round 2

Reviewer 1 Report

The author have made some important corrections which indeed led to a change in the configurational assignment of the new compound. The reason for me to ask for a more careful and quantitative evaluation of  crossrelaxation data was driven exactly by the possibility for a misassignment which even today happens still too often. I'd like to encourage the authors to think about using techniques like floating chirality distance geometry calculations using quantitative distance data. Hoping for a correct assignment now, I agree publication.

Author Response

Dear reviewer,

Thank you so much for your carefully reviewing work for our manuscript. We feel that our manuscript has been improved so much after the two revisions. Please check our response as to your comments as follows.

The author have made some important corrections which indeed led to a change in the configurational assignment of the new compound. The reason for me to ask for a more careful and quantitative evaluation of  crossrelaxation data was driven exactly by the possibility for a misassignment which even today happens still too often. I'd like to encourage the authors to think about using techniques like floating chirality distance geometry calculations using quantitative distance data. Hoping for a correct assignment now, I agree publication.

Response: 

Thanks for your suggestion on the NMR measurement, which enabled us to know a quantitative NMR method. We will try to learn the new method for determining the configurations of more complex structures. We determined ROESY and 1D NMR for analyzing the relative configuration of compound 1 that could solve the chiral carbons in the structure. 

Sincerely,

Reviewer 2 Report

Dear Authors,

The manuscript reported one new natural product and ten known compounds from endophytic Fungus Xylaria arbuscula of plant Rauvolfia vomitoria collected at China. The structure was studied by 2D NMR, MS and DFT-ECD, and all isolated compounds were tested for their NO inhibitory and cytotoxic activities.  

Minor: Journal name "Chemistry - A European Journal" for ref [18] is missing. Author might briefly the conclusion of compound 1 as hybrid PKS-NRPS, or provide references of biosynthetic papers with similar references.

Author Response

Dear reviewer,

Thanks for your carefully reviewing work for our manuscript. We feel that our manuscript has been improved so much after the two revisions. Please check our response as to your comments as follows.

Your comments:

The manuscript reported one new natural product and ten known compounds from endophytic Fungus Xylaria arbuscula of plant Rauvolfia vomitoria collected at China. The structure was studied by 2D NMR, MS and DFT-ECD, and all isolated compounds were tested for their NO inhibitory and cytotoxic activities.  

Minor: Journal name "Chemistry - A European Journal" for ref [18] is missing. Author might briefly the conclusion of compound 1 as hybrid PKS-NRPS, or provide references of biosynthetic papers with similar references.

Our Response: Thank you for your comments. We have corrected the missing content in the ref [18], and supplemented the biosynthesis introductions of the analogues of 1 and the corresponding reference in the manusctipt. We have marked the revised places by yellow highlights.